# Recycling of Coal Ash in Concrete as a Partial Cementitious Resource

**Sajjad Ali Mangi [1,2]**, **Mohd Haziman Wan Ibrahim [1,\*]**, **Norwati Jamaluddin [1]**, **Mohd Fadzil Arshad [3]** and **Sri Wiwoho Mudjanarko [4]**

[1]  Jamilus Research Center, Faculty of Civil and Environmental Engineering, Universiti Tun Hussein Onn Malaysia, Parit Raja 86400, Johor, Malaysia; sajjad.nec@gmail.com (S.A.M.); norwati@uthm.edu.my (N.J.)

[2]  Mehran University of Engineering and Technology, SZAB Campus, Khairpur Mirs 66020, Sindh, Pakistan

[3]  Faculty of Civil Engineering, Universiti Teknologi MARA, Shah Alam 40450, Selangor, Malaysia; mohdfadzil.arshad@salam.uitm.edu.my

[4]  Faculty of Engineering, Noratama University, Surabaya 60117, Indonesia; sri.wiwoho@narotama.ac.id

\*  Correspondence: haziman@uthm.edu.my; Tel.: +6074564202

**Abstract:** Concrete construction offers a great opportunity to replace the cement with a coal-based power plant waste—known as coal bottom ash (CBA)—which offers great environmental and technical benefits. These are significant in sustainable concrete construction. This study aims to recycle CBA in concrete and evaluate its particle fineness influence on workability, compressive and tensile strength of concrete. In this study, a total of 120 specimens were prepared, in which ground CBA with a different fineness was used as a partial cement replacement of 0% to 30% the weight of cement. It was noticed that workability was decreased due to an increased amount of ground CBA, because it absorbed more water in the concrete mix. The growth in the compressive and tensile strength of concrete with ground CBA was not significant at the early ages. At 28 days, a targeted compressive strength of 35 MPa was achieved with the 10% ground CBA. However, it required a longer time to achieve a 44.5 MPa strength of control mix. This shows that the pozzolanic reaction was not initiated up to 28 days. It was experimentally explored that 10% ground CBA—having particle fineness around 65% to 75% and passed through 63 μm sieve—could achieve the adequate compressive and tensile strength of concrete. This study confirmed that the particle fineness of cement replacement materials has a significant influence on strength performance of concrete.

**Keywords:** coal bottom ash; particle fineness; workability; compressive strength; tensile strength

## 1. Introduction

The thermal power plants operated on coal as an energy source produce two types of waste products, one is coal fly ash (CFA) and another is coal bottom ash (CBA). However, the light form of coal ash that floats into the exhaust stacks is known as CFA and the heavier portion of coal ash that settles on the ground in the boiler is recognized as CBA. According to Ahmaruzzaman [1], it was estimated that the production of coal ash is around 600 million tons worldwide per year, with CFA ash constituting around 500 million. The fly ash has similar properties to cement and has been adopted as a supplementary cementitious substance in concrete, but CBA is not frequently utilized in any form [2]. The current disposal practice of CBA in ponds poses a high risk to human health and the environment [2,3]. The utilization of CBA is still limited due to its comparatively higher content of unburned carbon and diverse structural features, compared to fly ash [4]. Considering the environmental benefits, the reuse of industrial waste in concrete production is the best alternative [5,6]. The particle size of CBA is large and similar to fine and coarse aggregates, therefore, previously researchers [2,5,7–9] considered CBA

as a fine aggregate replacement in concrete. CBA has pozzolanic characteristics and could potentially be utilized in concrete as a cement replacement material, by reducing its particle size.

A recent study was conducted on the use of ground CBA as cement constituents in concrete by Argiz, et al. [10]. This study compared the performances of CFA ash and ground CBA, in terms of strength and durability, and found that ground CBA had durability. But they did not consider the different fineness (different grinding periods). Moreso, it was previously suggested by Zhao, et al. [11], Jaya, et al. [12] and Mangi, et al. [13] that the addition of fine ground supplementary cementing materials (SCMs) could enhance the strength of concrete. It was previously stated that SCMs—made from natural or waste materials—has been given serious attention to the development of composite cement for ecological, economic, and multi-quality reasons [14]. However, the previous research indicates that the finer the particle size of the SCM, the more its hydration activity and rate of hydration [12]. To make better use of CBA, the mechanical grinding method can be utilized to get a finer particle size. Therefore, this study evaluated the ground CBA particle fineness effects on the concrete properties.

## 2. Materials and Methods

### 2.1. Materials

The ordinary Portland cement (OPC) Type-I was used as main binding material along with coarse aggregate passed through a 10 mm sieve and retained on a 5 mm sieve, as shown in Figure 1a. In addition, fine aggregate (sand) was passed through a 5 mm sieve, as shown in Figure 1b. The raw CBA, as shown in Figure 1c, was collected from Kapar Energy Vetures, Selangor, Malaysia.

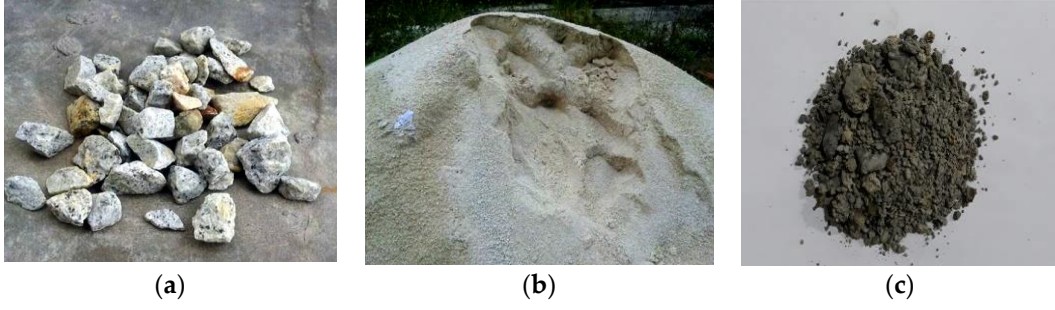

(**a**)   (**b**)   (**c**)

**Figure 1.** (**a**) Coarse aggregate, (**b**) fine aggregate, (**c**) raw coal bottom ash (CBA).

The raw CBA was dried in an oven at 110 ± 5 °C for 24 h. Afterward, it was placed in a Los Angeles machine for 2 h of initial grinding. Next, ground CBA was passed from a 300 micron sieve and placed into a ball mill grinder. The ball mill grinder produced three types of particle sizes: Type A, B and C (through three different grinding periods: 20, 30 and 40 h, respectively). The particle fineness was prepared in line with the American Society for Testing and Materials (ASTM) C618 [15] and graphically presented in Figure 2, which shows that the fineness of all three types of CBA. Type A, Type B and Type C are very close to the fineness of OPC.

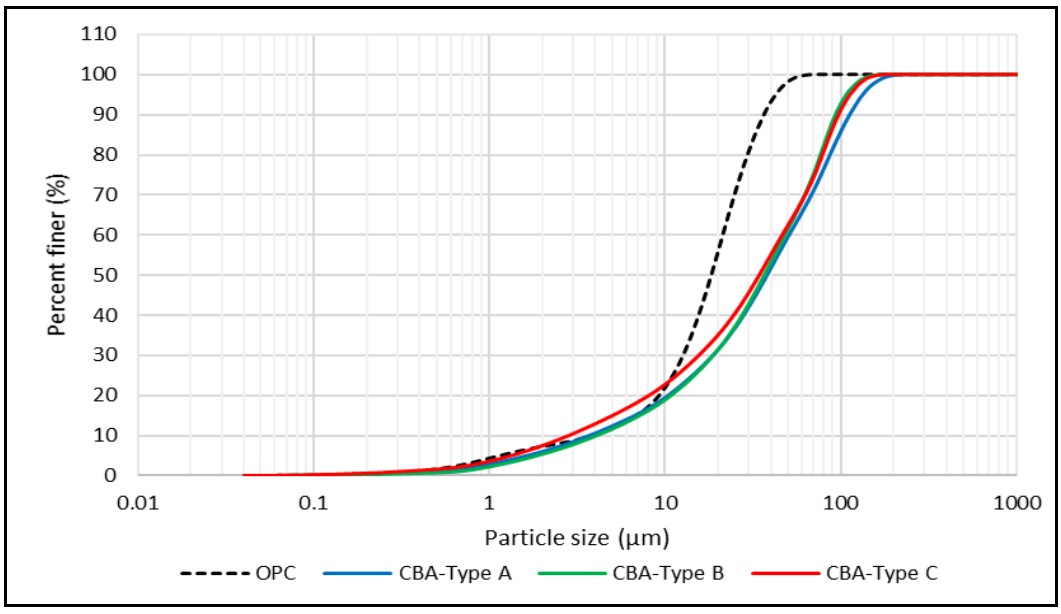

**Figure 2.** Particle fineness of ground CBA and ordinary Portland cement (OPC).

## 2.2. Physical Properties of CBA

The physical properties, such as the particle fineness, was evaluated with a particle size analyzer (PSA) instrument and the specific gravity was determined with Pycnometer test. However, PSA has the capacity to measure particle ranges from 0.04 to 2500 μm, it is a very fast and accurate method to measure the particle fineness in an acquisition time ≤1 min. It was observed that the original CBA had grey color and due to the grinding process, it changed from grey to dark grey in color and the specific gravity also varies. In this study, CBA of three different fineness properties were used, because particle fineness has an important role in the development of pozzolanic reaction [11]. In addition to particle size analysis, manual sieving was also performed through a 63 micron sieve to get the different particle fineness of the ground CBA. The physical properties of ground CBA and ordinary Portland cement (OPC) were evaluated and the results of specific surface area, specific gravity and percentage passing from sieve 63 micron are provided in Table 1.

**Table 1.** Physical properties of ground CBA and OPC.

| Properties | OPC | CBA with Different Fineness | | |
| --- | --- | --- | --- | --- |
| | | Type—A | Type—B | Type—C |
| Specific surface area (m$^2$/kg) | 487.081 | 383.575 | 389.484 | 463.778 |
| Specific gravity | 3.10 | 2.41 | 2.44 | 2.50 |
| Passing (%) from sieve 63 μm | 100 | 65.47 | 75.40 | 85.73 |
| Color | Grey | Grey | Dark grey | Dark grey |
| Grinding period (h) | - | 20 | 30 | 40 |

## 2.3. Chemical Composition of CBA

An X-ray fluorescence (XRF) machine produced by Bruker AXS S4 pioneer model from Germany was used to determine the chemical composition of the CBA and OPC as per ASTM E1621-13 [16]. The was sample prepared with 8 g of material and 2 g powder of wax. It was found that the CBA contained compounds of silicates, aluminates and iron oxide, with other compounds in minor fractions. The results of the chemical composition analysis of OPC and ground CBA are provided in Table 2. The chemical composition of the CBA at the different fineness was evaluated and all ashes were found to be Class-F pozzolanic materials, as per ASTM C618 [15]. However, silica, alumina and ferric are the important compounds that characterize the pozzolanic property of the material [17]. According to

Table 2 results, the sum of components $SiO_2$, $Al_2O_3$, and $Fe_2O_3$ was more than 70%. The LOI and $SO_3$ amount was not more than 5% and 6%, respectively.

**Table 2.** Chemical composition of OPC and CBA with different fineness.

| Sample | OPC | CBA |
|---|---|---|
| **Chemical Content** | **%** | **%** |
| Silica dioxide ($SiO_2$) | 20.61 | 53.80 |
| Aluminum Trioxide ($Al_2O_3$) | 3.95 | 18.10 |
| Ferric Oxide ($Fe_2O_3$) | 3.46 | 8.70 |
| Calcium Oxide (CaO) | 63.95 | 5.30 |
| Titanium dioxide ($TiO_2$) | 0.20 | 1.20 |
| Carbon (C) | - | 0.10 |
| Potassium Oxide ($K_2O$) | - | 0.85 |
| Magnesia/Magnesium Oxide (MgO) | 1.93 | 0.58 |
| Strontium oxide (SrO) | - | 0.35 |
| Phosphorus pentoxide ($P_2O_5$) | - | 0.29 |
| Sulfur trioxide ($SO_3$) | 3.62 | 0.90 |
| Barium oxide (BaO) | - | 0.18 |
| Zirconium dioxide ($ZrO_2$) | - | 0.15 |
| Sodium superoxide ($Na_2O$) | - | 0.17 |
| Loss on ignition (LOI) | 2.18 | 4.02 |

## 2.4. Concrete Mix Proportion

A total of ten concrete mixes were prepared with a 0.5 water to binder ratio. The first mix was prepared as a control mix (without ground CBA) and other nine contained ground CBA as a supplementary cementing material with different fine particle sizes and with different proportions. The American concrete institute (ACI) method of concrete mix was used and the details of the mix design are provided in Table 3.

**Table 3.** Detail of concrete mix proportions ($kg/m^3$).

| Notation | CBA Type | % Replacement | Cement | Ground CBA | Sand | Coarse Aggregate | Water |
|---|---|---|---|---|---|---|---|
| CM | - | 0 | 440 | 0 | 805 | 828 | 220 |
| M1 | | 10 | 396 | 44 | 805 | 828 | 220 |
| M2 | A | 20 | 352 | 88 | 805 | 828 | 220 |
| M3 | | 30 | 308 | 132 | 805 | 828 | 220 |
| M4 | | 10 | 396 | 44 | 805 | 828 | 220 |
| M5 | B | 20 | 352 | 88 | 805 | 828 | 220 |
| M6 | | 30 | 308 | 132 | 805 | 828 | 220 |
| M7 | | 10 | 396 | 44 | 805 | 828 | 220 |
| M8 | C | 20 | 352 | 88 | 805 | 828 | 220 |
| M9 | | 30 | 308 | 132 | 805 | 828 | 220 |

## 2.5. Sample Preparation

The procedures described in the American Standard ASTM C192/C 192M [18] were adopted for mixing, casting and curing of concrete. Potable tap water was used for concrete mixing and curing [19]. A mini drum mixer was used for the concrete mixing, and after mixing, cubical molds of size 100 mm × 100 mm were cast for the evaluation of compressive strength. Cylindrical molds 100 mm in diameter and 200 mm in length were cast for the evaluation of tensile strength. The de-molding was performed after one day of casting. Afterward, specimens were placed in a water tank under the submerged

condition at room temperature. A total of 120 specimens were prepared for this study. The details of the samples are provided in Table 4.

**Table 4.** Detail of the specimens.

| Sample Code | % Replacement | Compressive Strength | | Splitting Tensile Strength | | Total Specimens |
|:-----------:|:-------------:|:--------------------:|:--:|:------------------------:|:--:|:---------------:|
| Curing Period (days) | | 7 | 28 | 7 | 28 | |
| CM | 0 | 3 | 3 | 3 | 3 | |
| M1 | 10 | 3 | 3 | 3 | 3 | |
| M2 | 20 | 3 | 3 | 3 | 3 | |
| M3 | 30 | 3 | 3 | 3 | 3 | |
| M4 | 10 | 3 | 3 | 3 | 3 | 120 |
| M5 | 20 | 3 | 3 | 3 | 3 | |
| M6 | 30 | 3 | 3 | 3 | 3 | |
| M7 | 10 | 3 | 3 | 3 | 3 | |
| M8 | 20 | 3 | 3 | 3 | 3 | |
| M9 | 30 | 3 | 3 | 3 | 3 | |

## 3. Results and Discussion

### 3.1. Workability

Fresh-state concrete was first assessed in the workability test, which measures through the slump test. A fixed water to binder ratio as 0.5 was adopted for all mixes. However, the slump test was conducted with reference to ASTM C143/C143M [20] to determine the workability of the fresh concrete containing the ground CBA. Workability is considered as an early stage property of freshly mixed concrete [21], which indicates its ability to be mixed, transported and placed with the least loss of consistency [22]. A total of ten concrete mixes were prepared and tested for the workability of the different batches and the targeted slump value was fixed to 50 ± 5 mm. The site experiment and slump test results for the different concrete mixes are provided in Figure 3. Referring to the experimental findings, it could be observed that the control mix (CM) showed the highest slump, which was 56 mm, while the M3, M6 and M9 mixtures showed the lowest slump, which were 35, 25 and 25 mm, respectively. However, M3 had higher workability than M6 and M9, because M3 contained CBA, having less surface area, which absorbed less water as compared to that of M6 and M9. Furthermore, it could be observed that the slump decreased as the replacement of ground CBA increased. The decreases of workability with the increase in levels of cement replacement may be due to the increase in the fineness of the ground CBA. Therefore, more water was absorbed in the mixture. It was also observed by Kurama and Kaya [22], and Khan and Ganesh [23], that the workability reduced as the percentage of ground CBA increased in the concrete mix. It was also observed by Rafieizonooz et al. [24] that slump values deceased as replacement level of the fly ash and bottom ash increases. It was experimentally determined that concrete containing 10% ground CBA could meet the required workability, as M1, M4 and M7 gave satisfactory slump values of 50, 48 and 45 mm, respectively. Overall, it was observed from the current and previous studies that workability decreases as the ground CBA percentage increases as compared to CM [25]. This occurs due to the absorption of more water by the fine particles of the ground CBA [23].

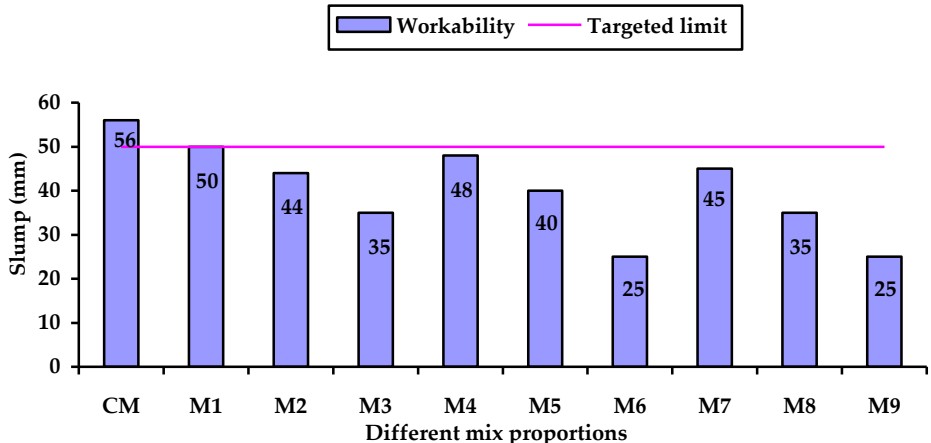

**Figure 3.** Slump test results of concrete with and without CBA and site experiment.

*3.2. Compressive Strength*

The universal load testing machine with a capacity of 3000 kN was used to evaluate the compressive strength of concrete. For this test, 100 mm cubical specimens were prepared. Three different fineness and three different proportions (10%, 20%, and 30%) were used to find out the optimum, based on 28 days performance. The test results of compressive strength are provided in Figure 4. A set of three cubes were cast for each mix proportion for each curing regime of 7 and 28 days and after the required curing time, the specimens were tested at a loading rate of 7 kN/s. The maximum load was recorded at failure and the average compressive strength was calculated.

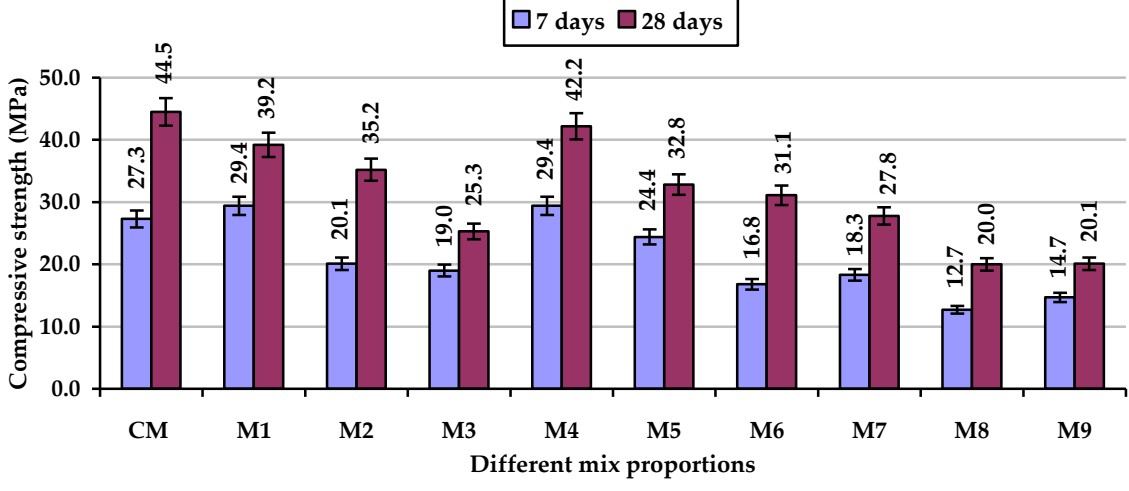

**Figure 4.** Concrete compressive strength with and without ground CBA.

From the experimental results as provided in Figure 4, it was noticed that the compressive strength of concrete incorporating ground CBA was lower than control mix. At an initial 7 days of curing, compressive strength of the concrete was found to increase with a 10% incorporation of ground CBA in M1, which showed the filler effect offered by ground CBA. Subsequently, at later curing days, the compressive strength of all mixes was lower than that of the control mix concrete. This was also reported by Rafieizonooz et al. [24] with the usage of fly ash as a replacement of cement in concrete. However, at up to 28 days, the pozzolanic reaction had not initiated significantly, so the early strength was due to the presence of cement. Moreover, Sena da Fonseca et al. [26] also declared that utilization of petrochemical waste as a cement replacement at 12% resulted in a 7.0% and 4.85% strength reduction at 28 and 90 days, respectively. However, this study considered coal-based power plant waste as a cementitious resource, with different fineness. Therefore, the prime results were observed with 10%

and 20% ground CBA of type A (M1 and M2) and 10% ground CBA of type B (M4) at a curing period 28 days, which achieved the targeted strength of 35 MPa. However, the performance of 10% ground CBA type A and B gave better results than the CBA type C, because finer particles provide a higher surface area and absorbed more available water in the mix, which creates an interruption in hydration process of the concrete mix. The strength of the control mix (CM) was higher than all the other mixes because more cement was present in the control mix as compared to other mixes [23,24]. Due to the addition of CBA, the hydration process of the cement-concrete was affected and it required more time to attain the strength of the control mix. In this study, it was experimentally determined that a 10% replacement of OPC with ground CBA is acceptable, which provided satisfactory compressive strength performances. These findings are supported by previous studies [22–24,27], which observed that concrete containing 10% ground CBA as a cement replacement was likewise found to be optimum.

### 3.3. Splitting Tensile Strength

Cylindrical specimens having a diameter of 100 mm and length of 200 mm were used for the determination of splitting tensile strength. However, three different proportions were used to replace the cement at 10%, 20%, and 30% for each fineness, to get the optimum proportion. This test was performed at 7 and 28 days. The results of tensile strength are provided in Figure 5. The maximum load was recorded at the appearance of a failure crack and the average tensile strength was calculated. It was observed through the experimental investigations that at an earlier age, there was no development in tensile strength of the concrete due to the addition of ground CBA. It can be seen from the 7-day results, that the tensile strength decreased as the amount of ground CBA increased. Subsequently, at 28 days, the tensile strength of the CBA-concrete was also lower than the control mix (CM), because the pozzolanic reaction was not yet initiated at this time point. Previously, Kurama and Kaya [22] declared that incorporation of 10% bottom ash delivers an adequate tensile strength at an age of 56 days. However, it is hereby declared that 10% CBA-Type B, as a replacement of OPC, is the optimum proportion based on the tensile strength performance.

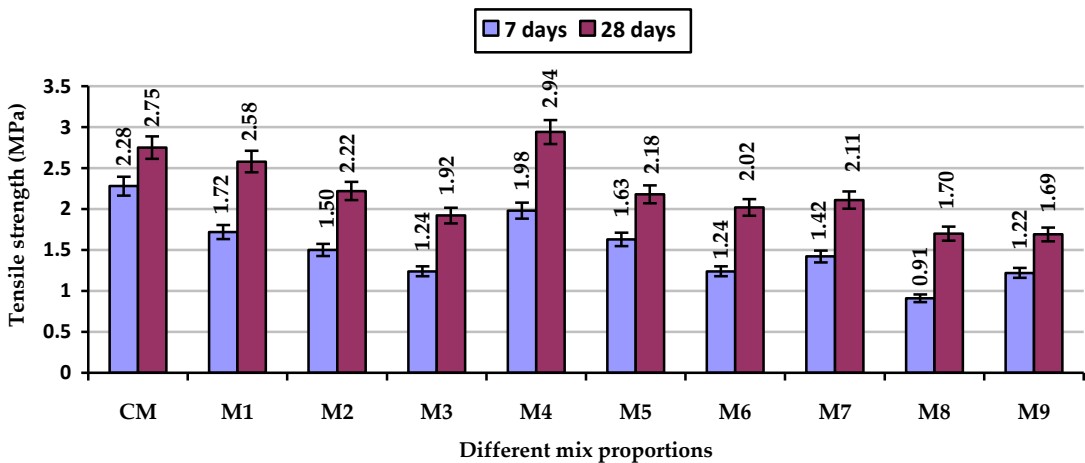

**Figure 5.** Concrete splitting tensile strength with and without ground CBA.

### 3.4. Compressive and Tensile Strength Relationship

After experimental findings, the linear strength relationship was determined through statistical procedures to assess the compressive and tensile strength behavior for the concrete containing ground CBA, with a different fineness and varying proportions, at an age of 28 days. In this study, the basic scientific theory of coefficients was adopted and the $R^2$ value was considered as the relationship coefficient, which was calculated through a regression curve analysis. The regression curve line shows the relationship between the independent variable (x-axis) and the dependent variable (y-axis) in the

graphs, as shown in Figure 6. The equation shows the association between compressive and tensile strength with the $R^2$ coefficients as provided below:

$$f_t = 0.0473\ f_{cu} + 0.7052 \qquad R^2 = 0.9235 \tag{1}$$

where:

$f_{cu}$: The compressive strength (MPa)
$f_t$: The split tensile strength (MPa)

However, the coefficient of $R^2$ obtained through a regression linear relationship was found to be comparable with the previous findings of Rafieizonooz et al. [24], who also found a comparable relationship in the concrete containing CBA and fly ash, as a sand and cement replacement, respectively.

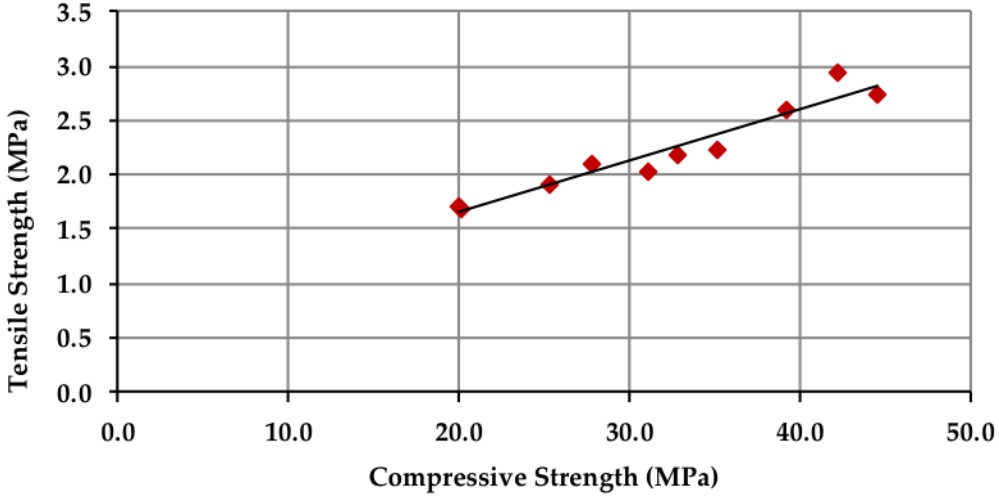

**Figure 6.** Compressive and tensile strength relationship.

## 4. Conclusions

This study highlighted the potential of CBA for actual operations. We demonstrate that CBA has good potential to be utilized as a cementitious resource. Moreover, the influence of CBA fineness on the concrete properties was also investigated. However, it was observed that the grinding process did not have a significant influence on the chemical composition of CBA, but had a greater influence on physical properties, such as the particle size and specific gravity. The results of this study reveal that a longer grinding period, greater fineness and greater specific gravity are appropriate. In addition, the following conclusions have been made:

- In this study, three types of ground CBA were produced: A, B and C (through three different grinding periods: 20, 30 and 40 h, respectively). It was experimentally demonstrated that CBA type A and B had good and satisfactory strength performance as compared to the CBA type C.
- The workability of the concrete was found to decrease with the increase in the amount of ground CBA as a cement replacement. It was noticed that the increase in the fineness of the ground CBA particles absorbed more water in the mixture.
- It was experimentally observed that concrete compressive strength was not developed with ground CBA at early ages. The compressive strength reduced with the addition of ground CBA as a cement replacement. Considering the results of different mixes, the optimum results were recorded with 10% of ground CBA in M1 and M4 at curing periods of 28 days, when both mixes achieved the targeted strength of 35 MPa.
- Similarly, the tensile strength of the concrete also reduced as the amount of ground CBA increased in the concrete mix, because the pozzolanic reaction was not yet initiated at an age of 28 days.

- It was experimentally explored that 10% ground CBA, having a particle fineness around 65% to 75%, passed through a 63 μm sieve, could achieve the adequate compressive and tensile strength of concrete.

Furthermore, considering the importance of ground CBA, it is hereby recommended for future research to prolong investigation on its strength performance during seawater exposure.

**Author Contributions:** Experimental investigation, analysis and writing—original draft preparation, S.A.M.; conceptualization, methodology, resources and supervision, M.H.W.I.; validation, N.J.; review, M.F.A.; editing, S.W.M.

**Funding:** This research was funded by the Universiti Tun Hussein Onn Malaysia and Ministry of Education Malaysia through Fundamental Research Grant Scheme (FRGS) Vot. No. K054. The support of Mehran University of Engineering and Technology, Pakistan in terms of FDP scholarship is also appreciated.

**Conflicts of Interest:** Authors declare no conflict of interest.

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
