# Peer review of "Recycling of Coal Ash in Concrete as a Partial Cementitious Resource"

_resources, doi:10.3390/resources8020099_

Round 1

Reviewer 1 Report

The paper aimed to investigate the effect of different types of coal bottom ash (CBA) addition on the concrete properties. The topic is interesting but the manuscript needs a revision to reach the standard level for publication.

Line 21-22: please include the results of reference (control) samples too.

Line 41: please merge all citations.

Lin 44: there is no need to have a comma before reference number 10.

Line 45: the authors stated “But he never considered …”. They did not consider …

Line 46-48: needs revision.

Please merge Figs 1-3 and label them as fig.1a-c

Please add reference(s) for all protocols used in the study.

Fig 5 and 6 can be removed.

What was the XRF model and working conditions during testing?

Line 93-95 can be removed.

Line 100: Have you previously introduced the full name of ACI acronym?

Figs 7 and 8 can be removed.

What was the curing conditions? Temperature?

Line 131: the authors stated that “This results were supported by the finding from Kurama & Kaya [21] and Khan & Ganesh [22] they were also found that the workability reduced as the percentage of ground CBA increases in the concrete mix.” This means the test has been already done and there is no novelty in this part.

Section 2.3: this section lacks from a comprehensive discussion on why and how the addition of CBA or any additives can affect the workability. For example why samples M3 and M9 had different workability (but they have same amount of cements, aggregates and …)

How many duplicates the authors used for compressive testing? The error bars must be added to Fig. 10 to see whether the data is significantly different or not.

As Fig. 10 shows, all samples had a lower strength than control. Also the increases in the replacement resulted a lower strength. The authors need to elaborate the reasons more.

Line 169: merge the citations. Same comment in line 171. Please check throughout the manuscript.

Line 184: please include the potential reasons for this contradiction.

Line 211: the authors stated: There was no significant development in concrete compressive strength with ground”. When you say “significant” it means you should have done statistical analysis (ANOVA). Have you done such a test?

Line 21-22: The authors stated “the target compressive strength of 35 MPa was achieved with 10% ground CBA”. But with a same mix (without CBA) a compressive strength of 44.5 MPa was attained which is significantly higher than the target strength.

Author Response

Thank you very much for your kind comments. All suggestions have been considered carefully and following changes have been made and changed highlighted in red type color on the attached revised manuscript.

S.No

Comments

Revision/Changes

1.          

Line 21-22:   please include the results of reference (control) samples too.

Result of control mix has been added

2.          

Line 41:   please merge all citations.

Citations have been merged

3.          

Lin 44: there   is no need to have a comma before reference number 10.

Comma has been removed

4.          

Line 45: the   authors stated “But he never considered …”. They did not consider …

Changes have been made as suggested

5.          

Line 46-48:   needs revision.

Revision has been made

6.          

Please merge   Figs 1-3 and label them as fig.1a-c

Fig.1-3 have been merged as suggested

7.          

Please add   reference(s) for all protocols used in the study.

References have been added

8.          

Fig 5 and 6   can be removed.

Removed

9.          

What was the   XRF model and working conditions during testing?

XRF model and its condition has been added

10.       

Line 93-95 can   be removed.

Removed

11.       

Line 100: Have   you previously introduced the full name of ACI acronym?

Full form of ACI has been added

12.       

Figs 7 and 8   can be removed.

Removed

13.       

What was the   curing conditions? Temperature?

Mentioned in the revised manuscript  

14.       

Line 131: the   authors stated that “This results were supported by the finding from Kurama   & Kaya [21] and Khan & Ganesh [22] they were also found that the   workability reduced as the percentage of ground CBA increases in the concrete   mix.” This means the test has been already done and there is no novelty in   this part.

Sentence has been revised

15.       

Section 2.3:   this section lacks from a comprehensive discussion on why and how the   addition of CBA or any additives can affect the workability. For example why   samples M3 and M9 had different workability (but they have same amount of   cements, aggregates and …)

workability is discussed section 3.1,   additional discussion has been added to support the findings.

16.       

How many   duplicates the authors used for compressive testing? The error bars must be   added to Fig. 10 to see whether the data is significantly different or not.

Error bars have been added

17.       

As Fig. 10   shows, all samples had a lower strength than control. Also the increases in   the replacement resulted a lower strength. The authors need to elaborate the   reasons more.

Reason has been incorporated

18.       

Line 169:   merge the citations. Same comment in line 171. Please check throughout the   manuscript.

Citations have been merged

19.       

Line 184:   please include the potential reasons for this contradiction.

Reason has been added.

20.       

Line 211: the authors   stated: There was no significant development in concrete compressive strength   with ground”. When you say “significant” it means you should have done   statistical analysis (ANOVA). Have you done such a test?

Sentence has been revised and significant word   has been deleted

21.       

Line 21-22:   The authors stated “the target compressive strength of 35 MPa was achieved   with 10% ground CBA”. But with a same mix (without CBA) a compressive   strength of 44.5 MPa was attained which is significantly higher than the target   strength.

Sentence has been modified and highlighted in   red type color

Reviewer 2 Report

The Topic of this study is relevant and suitable for publication in Resources.

1. Line 31. What the differences between coal fly ash and coal bottom ash? It is necessary to add more information to the Introduction section.

2. Line 37. …High content… What is this value?

3. Table 1. Fineness (cm2 /g) is not correct. It must be changed to a specific surface area (m2 /g).

4. Figures 5, 6 Can be removed. These figures are not added the new scientific information. It's necessary to include the name and country of equipment in the Materials section.

5. Line 92. Changed SiO3 to SiO2.

6. Line 190. Add space between 28days.

7. Figure 12. It makes Excel software. It can remake it in OriginPro. Now, these Figure terrible quality.

8. Conclusions. Please make this item first in conclusions: In this study three types of ground CBA was produced; A, B and C through three different grinding periods; 20, 30 and 40 hours respectively. It was experimentally demonstrated that CBA type A and B have good and satisfactory strength performance as compared to the CBA type C.

The main question to the Authors: In Figure 4 use a very long time to crush CBA. Why is this duration so long? Is it better to use 1-3 hours?

If the specific surface area of CBA has a significant effect on compressive strength and splitting tensile strength, you must describe more detail this information. In my opinion, you need to consider this when changing the title of the article. Add information about the particle size or specific surface area to the effect on the strength indicators of Portland cement.

This information provides to understand your article best to further readers.

Author Response

Thank you very much for your kind review comments. I have carefully considered all your suggestions and changes have been made in red type color on the attached revised manuscript.

A list of changes is also provided herewith.

S.No

Comments

Revision/Changes

1

 Line 31. What the   differences between coal fly ash and coal bottom ash? It is necessary to add   more information to the Introduction section.

Additional introduction has been added about   fly ash coal bottom ash.

2

Line 37…High   content… What is this value?

Value not know, different   ashes have different carbon content.

3

Table 1. Fineness   (cm2 /g) is not correct. It must be changed to a specific   surface area (m2 /g).

Replaced word fineness with specific surface   area in Table 1

4

Figures 5, 6 Can be removed.   These figures are not added the new scientific information. It's necessary to   include the name and country of equipment in the Materials section.

Removed

5

Line 92. Changed   SiO3 to SiO2.

Changed as suggested

6

Line 190. Add   space between 28days.

Modified as suggested

7

Figure 12. It makes   Excel software. It can remake it in OriginPro. Now, these Figure terrible   quality.

Figure quality has been improved

8

Conclusions. Please   make this item first in conclusions: In this study three types of   ground CBA was produced; A, B and C through three different grinding periods;   20, 30 and 40 hours respectively. It was experimentally demonstrated that CBA   type A and B have good and satisfactory strength performance as compared to   the CBA type C.

Changes have been made as suggested

9

The main   question to the Authors: In Figure 4 use a very long time to   crush CBA. Why is this duration so long? Is it better to use 1-3 hours?

Grinding of CBA was initiated with 2 hrs, 10,   hrs, 20 hrs, 30 hrs, 40 and 50 hrs. based on different trials, 20, 30 and 40   hrs have been selected for this study.

10

If the   specific surface area of CBA has a significant effect on compressive strength   and splitting tensile strength, you must describe more detail this   information. In my opinion, you need to consider this when changing the title   of the article. Add information about the particle size or specific surface   area to the effect on the strength indicators of Portland cement.

It was observed through this study that sp.   Surface (SA) of CBA has influence on performance of CBA because finer   particle absorbed more water in the mix, resulting in the decrease in   workability and strength. Therefore, appropriate grinding should be adopted   as to achieved targeted performances of the concrete.

Round 2

Reviewer 1 Report

The authors carefully addressed the comments and at this stage, I support the manuscript for publication.

Author Response

Thank you very much for your valuable review comments. All suggested changes have been carefully addressed in revised version and highlighted in red type color. Furthermore, figures 3 to 6 have been revised. 

Summary of changes:

S.No

Comments

Revision/Changes

1.          

Figure 1a, Fig 1b and Fig 1c must be changed   to Fig 1. ... a - , b - , c - .

Changes   has been made as suggested

2.          

Table 1. The specific surface area must be   marked by m^2/g, not cm^2/g. So, the data must be changed from 3835.75 to   3.835 and etc.

Unit of surface area has been revised as suggested

3.          

Figures 4, 5, 6. Become even worse quality —   the error bars imposed on the column values. There is no X, Y axis lines.   Figure 6 did not receive any significant changes at all. Changing the scale   along the X-axis is necessary.

Fig.   4, 5, 6, have been revised as suggested.

Reviewer 2 Report

First of all, thank you for your sincere response to my comments. Contents have improved a lot, but there are still some questions.

1. Figure 1a, Fig 1b and Fig 1c must be changed to Fig 1. ... a - , b - , c - .

2. Table 1. The specific surface area must be marked by m^2/g, not cm^2/g. So, the data must be changed from 3835.75 to 3.835 and etc.

3. Figures 4, 5, 6. Become even worse quality — the error bars imposed on the column values. There is no X, Y axis lines. Figure 6 did not receive any significant changes at all. Changing the scale along the X-axis is necessary.

Author Response

Thank you very much for your valuable review comments. All suggested changes have been carefully addressed in revised version and highlighted in red type color. Furthermore, figures 3 to 6 have been revised.

S.No

Comments

Revision/Changes

1.          

Figure 1a, Fig 1b and Fig 1c must be changed   to Fig 1. ... a - , b - , c - .

Changes has been made as suggested

2.          

Table 1. The specific surface area must be   marked by m^2/g, not cm^2/g. So, the data must be changed from 3835.75 to   3.835 and etc.

Unit of surface area has   been revised as suggested

3.          

Figures 4, 5, 6. Become even worse quality —   the error bars imposed on the column values. There is no X, Y axis lines.   Figure 6 did not receive any significant changes at all. Changing the scale   along the X-axis is necessary.

Fig. 4, 5, 6, have been revised as suggested.

Round 3

Reviewer 2 Report

The authors have done a lot of work on the errors. After reading a new version of the article, my opinion about it has changed, and I recommend this article for publication in the Resources in the present form.

Author Response

Thank you very much for your kind recommendation, consideration and encouragement.